# PRMT1 Regulates EGFR and Wnt Signaling Pathways and Is a Promising Target for Combinatorial Treatment of Breast Cancer

**DOI:** 10.3390/cancers14020306

**Published:** 2022-01-08

**Authors:** Samyuktha Suresh, Solène Huard, Amélie Brisson, Fariba Némati, Rayan Dakroub, Coralie Poulard, Mengliang Ye, Elise Martel, Cécile Reyes, David C. Silvestre, Didier Meseure, André Nicolas, David Gentien, Hussein Fayyad-Kazan, Muriel Le Romancer, Didier Decaudin, Sergio Roman-Roman, Thierry Dubois

**Affiliations:** 1Breast Cancer Biology Group, Translational Research Department, Institut Curie-PSL Research University, 75005 Paris, France; samyuktha.suresh@curie.fr (S.S.); solene.huard@curie.fr (S.H.); brisson.amelie@wanadoo.fr (A.B.); rayan.dakroub@curie.fr (R.D.); mengliang.ye@curie.fr (M.Y.); davidcsilvestre@gmail.com (D.C.S.); 2Pre-Clinical Investigation Laboratory, Translational Research Department, Institut Curie-PSL Research University, 75005 Paris, France; fariba.nemati@curie.fr (F.N.); didier.decaudin@curie.fr (D.D.); 3Laboratory of Cancer Biology and Molecular Immunology, Faculty of Sciences-I, Lebanese University, Hadath, Beirut 1003, Lebanon; hussein.kazan@ul.edu.lb; 4Cancer Research Center of Lyon, CNRS UMR5286, Inserm U1052, University of Lyon, 69000 Lyon, France; coralie.poulard@lyon.unicancer.fr (C.P.); muriel.leromancer-cherifi@lyon.unicancer.fr (M.L.R.); 5Platform of Experimental Pathology, Department of Diagnostic and Theranostic Medicine, Institut Curie-Hospital, 75005 Paris, France; elise.martel@curie.fr (E.M.); didier.meseure@curie.fr (D.M.); andre.nicolas@curie.fr (A.N.); 6Genomics Core Facility, Translational Research Department, Institut Curie-PSL Research University, 75005 Paris, France; cecile.reyes@curie.fr (C.R.); david.gentien@curie.fr (D.G.); 7Translational Research Department, Institut Curie-PSL Research University, 75005 Paris, France; sergio.roman-roman@curie.fr

**Keywords:** breast cancer, EGFR, PRMT1, drug combinations, Wnt signaling

## Abstract

**Simple Summary:**

Patients with triple-negative breast cancer (TNBC) respond well to chemotherapy initially but are prone to relapse. Searching for new therapeutic targets, we found that PRMT1 is highly expressed in TNBC tumor samples and is essential for breast cancer cell survival. Furthermore, this study proposes that targeting PRMT1 in combination with chemotherapies could improve the survival outcome of TNBC patients.

**Abstract:**

Identifying new therapeutic strategies for triple-negative breast cancer (TNBC) patients is a priority as these patients are highly prone to relapse after chemotherapy. Here, we found that protein arginine methyltransferase 1 (PRMT1) is highly expressed in all breast cancer subtypes. PRMT1 depletion decreases cell survival by inducing DNA damage and apoptosis in various breast cancer cell lines. Transcriptomic analysis and chromatin immunoprecipitation revealed that PRMT1 regulates the epidermal growth factor receptor (EGFR) and the Wnt signaling pathways, reported to be activated in TNBC. PRMT1 enzymatic activity is also required to stimulate the canonical Wnt pathway. Type I PRMT inhibitors decrease breast cancer cell proliferation and show anti-tumor activity in a TNBC xenograft model. These inhibitors display synergistic interactions with some chemotherapies used to treat TNBC patients as well as erlotinib, an EGFR inhibitor. Therefore, targeting PRMT1 in combination with these chemotherapies may improve existing treatments for TNBC patients.

## 1. Introduction

Breast cancer (BC) is a heterogeneous disease with molecularly distinct subtypes displaying different clinical outcomes and responses to therapies [1]. Patients with “triple-negative” breast cancer (TNBC, lacking the expression of estrogen and progesterone receptors and Her2 overexpression) are mainly treated with conventional chemotherapies [1,2]. However, these patients have the worst prognosis as their treatment is challenging, due to their inter- and intra-tumor heterogeneity, leading to resistance to chemotherapy and relapse [2]. Therefore, more efficacious treatments are needed to improve TNBC patient survival.

EGFR is overexpressed in more than 70% of TNBC patients and is associated with a metastatic phenotype [3]. However, targeting this receptor as a monotherapy has shown only modest to low efficacy in clinical trials for TNBC patients [3]. The Wnt signaling pathway is another pathway activated in TNBC through an overexpression of the transmembrane receptors, Frizzleds and co-receptors low-density lipoprotein receptor-related proteins (LRP6 and LRP5) [4,5,6]. Wnt ligands (such as Wnt3a), which are secreted upon palmitoylation by the enzyme porcupine [7], activate the Wnt pathway by binding to the transmembrane receptors Frizzleds and co-receptors LRP5/LRP6. This initiates the release of β-catenin from the destruction complex including Dishevelled and Axin. Free β-catenin translocates into the nucleus and binds to the TCF/LEF family of transcription factors to activate the expression of Wnt target genes [4,5,7].

Arginine methylation of histone and non-histone proteins is a post-translational modification catalyzed by Protein Arginine Methyltransferases (PRMTs) [8,9,10,11,12,13]. Substrate arginine can either be monomethylated or dimethylated (symmetrically or asymmetrically) by PRMTs. Type I PRMTs (PRMT1-4, PRMT6, and PRMT8) are responsible for asymmetric dimethylation and Type II PRMTs (PRMT5 and PRMT9) for symmetric dimethylation [8,9,10,11,12,13,14]. PRMTs are ubiquitously expressed, except PRMT8 which is brain-specific [12]. Several PRMTs are overexpressed in various cancer types, including breast cancer, [13,15,16,17] and are emerging as attractive therapeutic targets [9,10,12,15]. Specific inhibitors targeting PRMT5 are being evaluated in phase I clinical trials (NCT03573310, NCT03854227, NCT02783300) [10]. PRMT1-specific inhibitors are not yet available, nevertheless, two type I PRMT inhibitors (MS023, GSK3368715) have been developed showing more efficacy towards PRMT1, PRMT6 and PRMT8 [18,19]. GSK3368715 is currently in a phase I clinical trial for diffuse large B-cell lymphomas and solid cancers (NCT03666988).

Arginine methylation regulates several cellular processes including transcriptional regulation and signal transduction [8,11,12]. PRMT1 and PRMT5 regulate the EGFR signaling pathway by methylating EGFR (in colorectal and TNBC cells) [20,21,22], or by methylating histones on the EGFR promoter (in glioblastoma or colorectal cells) [23,24] to regulate its transcription. Furthermore, some PRMTs regulate the canonical Wnt signaling pathway [12]. Indeed, PRMT1 could either activate this pathway by methylating G3BP1 or G3BP2 [25,26], or inhibit it by methylating Axin and Dishevelled [27,28]. Whether PRMT1 regulates the Wnt pathway in breast cancer cells is still unknown.

PRMT1 has been mainly studied in luminal BC due to its well-described function as a transcriptional coactivator of estrogen receptor (ER) [15,29]. However, its implication in the other BC subtypes, specifically in TNBC, remains to be explored. Here, we examined the expression of PRMT1 in the different breast cancer subtypes, evaluated its potential as a therapeutic target, and explored its function in TNBC cells.

## 2. Materials and Methods

### 2.1. Human Samples

Our cohort has been previously described [6,17,30,31] and is composed of 35 luminal A (LA), 40 luminal B (LB), 46 TNBC, 33 Her2+, and 18 normal breast tissues. Experiments were conducted in accordance with Bioethics Law No. 2004–800 and the Ethics Charter from the French National Institute of Cancer (INCa), and after approval from the ethics committee of our Institution. DNA (Affymetrix SNP 6.0, ThermoFisher Scientific, Waltham, MA, USA) and RNA (Affymetrix U133 plus 2.0, ThermoFisher Scientific) microarrays on this cohort have been previously described [30].

### 2.2. Immunohistochemistry (IHC)

PRMT1 IHC was carried out on tissue microarrays (TMA), containing alcohol, formalin and acetic acid (AFA)-fixed paraffin-embedded tissues, as previously described [30], using a rabbit PRMT1 polyclonal antibody (Appendix A). This antibody recognizes residues 298–318 within the C-terminal domain of all isoforms of PRMT1. PRMT1 antibody was validated and optimized for IHC using AFA-fixed pellets from MDA-MB-468 cells treated with PRMT1 siRNAs or control siRNA for 72 h (Appendix A). For surface staining quantifications, whole digital slide images were obtained using virtual microscopy (Philips Ultra-Fast Scanner 1.6 RA, Amsterdam, Netherlands) and analyzed with Digital Image Analysis platform HALO (version 3.0.311.218; Indica Lab, Albuquerque, NM, USA). Tissue classifier was trained to segment tumor tissue and stroma. Area Quantification module (v2.1.3, Albuquerque, NM, USA) was used to evaluate the area of each tissue class and the area of tissue positive for PRMT1 staining. For subcellular localization of PRMT1 (plasma membrane, nucleus, and cytosol), TMA were read by two pathologists who assigned intensity scores (0–3) for each compartment (0: no staining, 3: strongest staining).

### 2.3. Cell Culture, RNA Interference, Antibodies, Small-Molecule Inhibitors, and Primers

TNBC (MDA-MB-468, HCC38, HCC70, MDA-MB-453), luminal (MCF7, T47D), and Her2+ (SKBr3, HCC1954, BT474) cell lines were purchased from the American Type Culture Collection (ATCC), authenticated in 2021 by short-tandem repeat profiling (data not shown), tested for mycoplasma using MycoAlert Mycoplasma Detection Kit (Lonza Biosciences, Durham, NC, USA), and cultured as previously described [17,32]. The murine cell lines, L-cells and L-Wnt3a were obtained from Institut de Recherches Servier, France. The MDA-MB-231 cell line was a kind gift from Dr. Mina Bissell (University of California, Berkeley, CA, USA). siRNA (20 nM) transfection was performed using Interferin (409-50, Polyplus, New York, NY, USA), according to the manufacturer’s instructions. References for antibodies, siRNAs, primers and drugs/small-molecule inhibitors are listed in Appendix A.

### 2.4. Cell Proliferation Assay

Cells were seeded in 96-well plates and cell proliferation was determined by MTT (M2128-1G, Sigma-Aldrich, St. Louis, MO, USA), WST-1 (11644807001, Sigma-Aldrich) or CellTiterGlo (G7572, Promega, Madison, WI, USA) assays as previously described [17,30,32].

### 2.5. Apoptosis Assays

Apoptotic activity was determined by the Caspase-Glo 3/7 luminescent assay (G8092, Promega), Annexin-V staining (11988549001, Roche, Basel, Switzerland) or Western blot analysis as previously described [17,30,32].

### 2.6. Colony Formation Assay

Cells transfected with siRNA were seeded in 6-well plates in 2 mL of growth media. Cells were incubated at 37 °C for 6 mitotic cycles (6–14 days), depending on the cell line, until colony formation. Colonies were fixed and stained with 500 µL of coomassie blue solution for 20 min. Colonies were photographed using a LAS-3000 Luminescent Image analyser (Fujifilm, Tokyo, Japan) or Chemidoc MP imager (Bio-rad Laboratories, Hercules, CA, USA) and quantified by ImageJ 1.43u software (NIH, Bethesda, MD, USA).

### 2.7. Soft Agar Assay

A 1 mL bottom layer consisting of 0.5% agar medium (equal volumes of 1% agar and 2 × culture medium) was added to 6-well plates. MDA-MB-468 cells were transfected with RNAi, and 24 h later, they were trypsinized, resuspended in 0.35% agar medium, and plated at 5000 cells/well as a top layer. Cells were incubated 4 weeks at 37 °C and the colonies were stained with an MTT assay. Plates were photographed with a Fujifilm LAS-3000 Imager, and the clones were quantified using Image J software.

### 2.8. Real-Time—Quantitative PCR Assay (RT-qPCR)

For Wnt target gene expression, MDA-MB-468 cells transfected with siRNA were serum-starved overnight and stimulated with Wnt3a conditioned media at 100 ng/mL for 6 h. RNA was extracted using the RNeasy Mini Kit (74106, Qiagen, Hilden, Germany) following the manufacturer’s protocol. Reverse-transcription and RT-qPCR were performed in a one-step reaction using the QuantiTect SYBR Green RT-PCR Kit (204245, Qiagen), according to the manufacturer’s protocol. The acquisition was made using a QuantStudio™ 12K Flex Real-Time PCR System (Applied Biosystems, Waltham, MA, USA).

### 2.9. β-Catenin-Activated Reporter (BAR) Luciferase Assay

At 24 h post siRNA transfection or 48 h post inhibitor treatment, MDA-MB-468 cells were transfected with the SuperTOPflash (7X Wnt response element containing plasmid) and pRL-TK-Renilla plasmids (both plasmids from Institut de Recherches Servier, Croissy-sur-Seine, France) at a 10:1 ratio using X-tremeGENE™ HP (6366236001, Sigma-Aldrich) as a transfectant. The cells were serum-starved overnight, i.e., 4–5 h post DNA transfection and stimulated with 100 ng/mL of Wnt3a conditioned media for 6 h. Dual-luciferase assay (E1910, Promega) was performed following manufacturer’s protocol, and the luminescence signal was measured on the Infinite M200 spectrophotometer (Tecan, Männedorf, Switzerland). The ratio of the signal from firefly (SuperTOPflashCroissy-sur-Seine, France) to renilla (pRL-TK-RenillaCroissy-sur-Seine, France) luciferase was calculated to obtain normalized luciferase activity, representing Wnt/β-catenin activity.

### 2.10. Chromatin Immunoprecipitation (ChIP)

Chromatin was prepared from 4 × 10^6^ untreated MDA-MB-468 cells using the simple ChIP plus enzymatic chromatin IP Kit (9004, Cell signaling Technology, Danvers, MA, USA), following the manufacturer’s protocol. The chromatin was immunoprecipitated using anti-PRMT1 or anti-IgG antibodies (Appendix A) overnight, and the chromatin/antibody complex was pulled down using protein G agarose beads (provided with the kit). Following different washing steps, the chromatin was eluted, and the cross links were reversed using proteinase K. DNA was purified using the spin columns included in the kit, and a qPCR was performed using specific primers designed based on a published ChIP-seq dataset for PRMT1 [33] for the promoter region of each gene (Appendix A).

### 2.11. Transcriptomic Analysis of PRMT1-Depleted Cells

The transcriptome of MDA-MB-468 cells depleted for PRMT1 was performed using Affymetrix HTA 2.0 microarray (ThermoFisher Scientific). Differential gene expression between control and PRMT1 siRNA with an adjusted *p*-value cut-off of 0.05 was considered (Appendix A). Gene enrichment pathway analysis was performed using the REACTOME database from the GSEA website [34].

### 2.12. GSK3368715 Treatment in Mice

Six-week-old female Swiss-nude mice were purchased from Charles River laboratories (Wilmington, MA, USA) and maintained in specific pathogen-free conditions. Their care and housing were per institutional guidelines as put forth by the French Ethical Committee. GSK3368715 (CS-0100240, ChemScene LLC, South Brunswick, NJ, USA) was formulated in 10% DMSO (Sigma-Aldrich) at 80 mg/mL and subsequently diluted in water. GSK3368715 toxicity studies were performed by administrating 100 mg/kg daily to nude mice.

MDA-MB-468 cells (12 × 10^6^ per mouse) were injected subcutaneously into nude mice until tumors reached 70 mm^3^. The tumor fragments obtained from 2 mice were then grafted into the inter-scapular fat pad of nude mice. Xenografts were randomly assigned to control or treatment groups (*n* = 6/group) when tumors reached a volume comprised between 60 and 80 mm^3^ and treated with vehicle or GSK3368715 at 80 mg/kg once daily orally 5 days/week. During the weekends, the inhibitor was added to the drinking water of mice. The tumor volume was evaluated by measuring two perpendicular tumor diameters with a caliper, twice a week. Mice were euthanized after 8 weeks of treatment. Tumor volumes were calculated as V = a × b^2^/2, a being the largest diameter, b the smallest. The tumor volumes were then reported to the initial volume as the relative tumor volume (RTV). Means of RTV in the same treatment group were calculated, and growth curves were established as a function of time.

### 2.13. Drug Combinations

MDA-MB-468 cells were seeded 48 h prior to treatment in a 96-well white transparent bottom plate (655098, Greiner Bio-One, Les Ulis, France) and treated with varying concentrations of the drugs/inhibitors. The maximum concentration for each drug/inhibitor was approximately twice the half maximal inhibitory concentration (2 × IC_50_) (Appendix A), and serially diluted two-fold for all drugs except for the type I PRMT inhibitors (three-fold). Cell viability was determined after 7 days of treatment by CellTiterGlo assay (G7572, Promega). The luminescence signal was measured in a Spark spectrophotometer (Tecan). Drug pair interactions using the Loewe model were calculated on the Combenefit software [35]. All drug combinations were performed in triplicate reactions per experiment.

### 2.14. Statistical Analysis

R software and GraphPad Prism 7 were used for statistical analyses. Pearson or Spearman correlation were used to estimate an association between two variables. For cellular assays, *p*-values were calculated using the Student *t*-test, unless otherwise specified. Independence between tumor subtypes in the TMA was assessed using Fisher’s exact test. 

All the whole western blot figures can be found in the Appendix A).

## 3. Results

### 3.1. PRMT1 Is Overexpressed in All the Breast Cancer Subtypes Compared to Normal Breast Tissue

With the goal of identifying enzymes overexpressed in BC compared to normal tissue, we have performed gene expression profiling on a cohort of 154 human BC biopsies and healthy breast tissues [6,17,30,31]. We found that *PRMT1* mRNA is overexpressed in all BC subtypes compared to normal tissues and observed the highest expression in TNBC (Figure 1A, left panel). The highest expression of *PRMT1* mRNA in TNBC was confirmed in the publicly available database—the cancer genome atlas (TCGA) cohort (Figure 1A, right panel). We examined whether variations in *PRMT1* expression could be a result of genomic alterations by analyzing DNA microarrays. Indeed, there was a correlation between *PRMT1* mRNA and the gene copy number within the whole cohort (Appendix A). Interestingly, the *PRMT1* locus showed significantly more gains in TNBC than the luminal BC subtypes and normal tissue (Figure 1B, Appendix A). The *PRMT1* mRNA levels also correlated positively with proliferation (*MKI67* mRNA) in our cohort (Appendix A).

To understand the clinical significance of *PRMT1* mRNA expression, we plotted survival outcomes from the KM-plotter database (Kaplan-Meier Plotter. Available online: https://kmplot.com/analysis/index.php?p=service&cancer=breast (accessed on 11 June 2021)) [36]. High *PRMT1* mRNA expression was associated with poor recurrence-free survival (RFS) in all BC (*p* = 1 × 10^−8^, Appendix A), as previously reported [37]. However, this analysis did not consider that *PRMT1* is differentially expressed among the BC subtypes (Figure 1A), which are associated with different prognoses. Therefore, we performed this analysis within the different BC subtypes. High *PRMT1* mRNA levels were associated with poor RFS in LA (*p* = 2.5 × 10^−6^) and LB (*p* = 0.007) (Appendix A, top panel). Although this trend was seen in the Her2+ subtype, it was not statistically significant (*p* = 0.13) (Appendix A, bottom left panel). Conversely, high *PRMT1* mRNA expression showed better RFS (*p* = 0.02) within the TNBC subtype (Appendix A, bottom right panel).

As mRNA and protein levels do not always coincide, we studied PRMT1 protein expression in breast tumors and normal tissues using a commercial PRMT1 antibody. We first validated this antibody for IHC staining in a TNBC cell line (MDA-MB-468) fixed in the same method as the tissue samples (Appendix A). IHC analysis confirmed that PRMT1 is highly expressed in all BC subtypes compared to normal tissues (Figure 1C,D). In contrast to mRNA expression, we did not observe any significant difference in PRMT1 protein expression levels between the different BC subtypes (Figure 1C,D). PRMT1 shows both nuclear and cytosolic staining (Figure 1C,E) and was also detected at the plasma membrane, mainly in ER-negative tumors (Figure 1E). Moreover, we observed substantial staining of PRMT1 in the stroma of breast tumors as compared to the normal tissues (Figure 1D). Mononuclear cells, fibroblasts and endothelial cells were positively stained for PRMT1 within the stroma (unpublished data).

Altogether, our results indicate that both PRMT1 mRNA and protein levels are higher in breast tumors compared to normal breast tissues, suggesting that PRMT1 could be targeted in BC.

### 3.2. RNAi-Mediated Depletion of PRMT1 Decreases BC Cell Viability, Clonogenicity and Induces DNA Damage and Apoptosis

To explore the function of PRMT1 in BC cells, we first depleted PRMT1 using two validated siRNAs (PRMT1#7, PRMT1#8) in MDA-MB-468 TNBC cells (Appendix A). We observed that cell viability was significantly decreased upon PRMT1 depletion in MDA-MB-468 cells, in a time-dependent manner (Figure 2A). Similar results were found in other BC cell lines (4 TNBC, 1 Her2+, 2 luminal; Appendix A), suggesting that the effect was independent of BC subtype. PRMT1 depletion decreased colony formation in MDA-MB-468 cells under adherent conditions (Figure 2B) or in an anchorage-independent growth assay in soft agar (Figure 2C), indicating that PRMT1 depletion decreases the tumorigenicity of this TNBC cell line. PRMT1 depletion also decreased colony formation in other BC cells cultured under adherent conditions (Appendix A). Furthermore, we observed a cleavage of caspases 3, 7, and PARP in MDA-MB-468 cells following PRMT1 depletion (Figure 2D), revealing apoptosis induction. This was confirmed in PRMT1-depleted MDA-MB-468 cells using a caspase 3/7 activity assay (Figure 2E) and by extracellular annexin-V staining (Figure 2F). PRMT1 depletion also significantly increased the phosphorylation of histone H2AX (γH2AX), a DNA damage marker (Figure 2D). The induction of apoptosis upon PRMT1 knockdown was confirmed in other BC cell lines (HCC70, MDA-MB-231, SKBr3, T47D; Appendix A). Together, these results demonstrate that PRMT1 is required for BC cell survival.

### 3.3. Type I PRMT Inhibitors Reduce BC Cell Growth

Next, we sought to explore if the enzymatic activity of PRMT1 was necessary for BC cell survival. For this purpose, we used two recently developed type I PRMT inhibitors: MS023 [18] and GSK3368715 [19]. Under the tested conditions, both inhibitors decreased the PRMT1-specific histone mark H4R3me2a without affecting the methylation of H3R17me2a (by CARM1 and PRMT6) or PABP1 (by CARM1; Appendix A). We tested the effect of both inhibitors on the cell viability in 5 TNBC (MDA-MB-468, MDA-MB-231, HCC38, HCC70, MDA-MB-453), 1 luminal (T47D) and 2 Her2+ (HCC1954, BT474) BC cell lines. HCC1954 cells were the most sensitive cells to type I PRMT inhibition (Figure 3A), followed by MDA-MB-468 and T47D cells (Figure 3A). The other TNBC cell lines were resistant to type I PRMT inhibition (IC_50_ > 10 µM, Figure 3A). We also observed smaller-sized colonies when MDA-MB-468 (Figure 3B) or four other TNBC cell lines (Appendix A) were treated with both inhibitors.

### 3.4. Type I PRMT Inhibition Slows Tumor Growth in a TNBC Xenograft Model

We evaluated the anti-tumor effect of inhibiting PRMT1 using GSK3368715, the only type I PRMT inhibitor currently in a phase I clinical trial for diffuse large B-cell lymphomas and solid tumors (NCT03666988). To better represent clinical conditions, we engrafted tumors derived from MDA-MB-468 cells into Swiss-nude mice (see Materials and Methods). GSK3368715 treatment significantly slowed tumor growth (*p* = 0.015; Figure 3C) with no observed toxicity (Appendix A). We confirmed that PRMT1 was indeed inhibited in the tumors at the end of the experiment by observing an increase in pan-monomethylation (Appendix A), as previously reported [38], and a decrease in histone H4R3 methylation (H4R3me2a, Figure 3D).

### 3.5. PRMT1 Regulates the EGFR and Wnt Signaling Pathways at the Transcriptomic Level

PRMT1 plays a crucial role in transcriptional regulation [8,11,12]. Therefore, we performed transcriptomic analysis of PRMT1 depleted MDA-MB-468 cells to gain insight into the molecular mechanisms that mediate the dependency of BC cells on PRMT1.

MDA-MB-468 cells were transfected with two different siRNAs targeting PRMT1 for 24 h and 48 h and the RNA were analyzed using HTA 2.0 microarrays (Affymetrix). We focused on the genes that were commonly deregulated at 24 h and 48 h by both siRNAs (Appendix A) to perform a gene enrichment pathway analysis using the REACTOME database [34]. The top ranked pathways (according to adjusted *p*-value) revealed that PRMT1 is involved in several cellular processes including signal transduction pathways, immune system response, lipid metabolism and transcriptional regulation (Appendix A). We focused on EGFR (*p* = 6.96 × 10^−6^) and Wnt (*p* = 5.07 × 10^−6^) signaling pathways, which are known to be activated in TNBC [3,4,5].

We noticed that *EGFR* mRNA itself was less expressed upon PRMT1 depletion in our microarray analysis (Figure 4A) and confirmed this observation by qPCR (Figure 4B). *EGFR* mRNA was also retrieved in several other deregulated pathways (Appendix A, arrowheads and diamond). PRMT1 was directly recruited to two promoter regions of *EGFR* in MDA-MB-468 cells using an anti-PRMT1 antibody (Figure 4C), previously validated for ChIP experiments [39]. Furthermore, PRMT1 depletion also decreased EGFR protein expression (Figure 4D).

Our microarray analysis revealed two key players of the Wnt signaling pathway, *LRP5* and *PORCN* (Porcupine), to be less expressed following PRMT1 depletion (Figure 4E). *LRP5* and *PORCN* mRNAs were also found in the second-top deregulated pathway (Appendix A, diamond). We validated the decrease in their expression by qPCR (Figure 4F) and identified by ChIP analysis that PRMT1 is enriched on the promoter of *LRP5* and two regions of the *PORCN* promoter (Figure 4G). The expression of LRP5 was also decreased at the protein level after PRMT1 depletion (Figure 4H). We could not assess porcupine protein expression due to the lack of suitable antibodies for Western blotting.

Overall, these results indicate that PRMT1 regulates the expression of *EGFR*, *LRP5* and *PORCN* by being recruited to their promoter regions.

### 3.6. PRMT1 Activates the Canonical Wnt Signaling Pathway

We hypothesized that PRMT1 could be an activator for the Wnt pathway as both LRP5 and PORCN are required for Wnt activation. We first assessed the Wnt activity by analyzing the expression of the three Wnt target genes (*AXIN2*, *APCDD1*, and *NKD1*) that are the most upregulated in Wnt3a-stimulated MDA-MB-468 cells [40]. We observed that PRMT1 depletion reduced the expression of these three Wnt target genes (Figure 5A). By using the gold standard β-catenin activated reporter (BAR) assay, we confirmed that PRMT1 depletion decreased Wnt signaling activity (Figure 5B). siRNA targeting LRP6 was used as a positive control in both assays (Figure 5A,B).

Next, we checked if PRMT1 enzymatic activity was involved in the regulation of Wnt pathway. MDA-MB-468 cells were treated for 3 days with low doses of MS023 or GSK3368715 (0.1 μM and 0.5 μM) and then stimulated for 6 h with Wnt3a, before assessing Wnt activity using the BAR assay (Figure 5C). Both type I PRMT inhibitors decreased the Wnt activity in a dose-dependent manner (Figure 5C). PRMT1 was inhibited under these conditions (Figure 5D).

Collectively, this demonstrates that PRMT1 and its activity are involved in the activation of the canonical Wnt pathway in MDA-MB-468 cells.

### 3.7. Type I PRMT Inhibitors Show Synergistic Interactions with Erlotinib or Chemotherapies

The rationale of drug combinations is to improve the efficacy, limit side-effects and reduce the risk of drug resistance. First, we combined both type I PRMT inhibitors with chemotherapies (cisplatin, camptothecin, cyclophosphamide, taxanes) used in the clinic to treat TNBC patients. MDA-MB-468 cells were treated with varying concentrations of the drugs, starting from about 2 × IC_50_ (Appendix A) for 7 days (equivalent to four mitotic cycles) and cell viability was assessed using CellTiterGlo assay. We applied the Loewe additivity model using the Combenefit software [35] to determine the nature (synergy/additivity/antagonism) of the drug interactions. We used this model as it allows the possibility to analyze two drugs that may act on the same pathway(s) [41]. Both type I PRMT inhibitors synergized with cisplatin (Figure 6A and Figure Appendix A), camptothecin (Figure 6B and Appendix A) and cyclophosphamide (Figure 6C and Appendix A), but not with docetaxel (Appendix A) or paclitaxel (Appendix A).

As EGFR is highly expressed in TNBC [3], we also evaluated the potential of combining type I PRMT inhibitors with an EGFR inhibitor (erlotinib) and observed a high synergy in MDA-MB-468 cells (Figure 6D and Appendix A). These combinations may represent promising alternative therapeutic strategies for TNBC patients.

## 4. Discussion

The efficacy of breast cancer therapies has considerably improved; however, TNBC still has a poor prognosis compared with other subtypes and is typically correlated with increased recurrence and worse survival. Finding alternative treatments to chemotherapy remains a priority to treat TNBC patients to avoid relapses.

PRMTs are overexpressed in various cancer types and are emerging as attractive therapeutic targets [8,9,10,11,12,13,14]. Consequently, several PRMT inhibitors have been developed, and some PRMT5 and type I PRMT inhibitors are being evaluated in clinical trials [10].

At the RNA level, we found that *PRMT1* is more expressed in BC when compared to the normal breast tissue, aligning with previous studies that did not consider BC heterogeneity [42,43]. *PRMT1* mRNA correlates positively with *MKI67* mRNA. Consequently, the highest *PRMT1* mRNA expression was found in TNBC, the most proliferative BC subtype, and this could be a result of DNA copy number gain. High *PRMT1* mRNA expression correlates with poor prognosis in all breast tumors, as reported in [37,44], as well as within LA and LB subtypes. In contrast, TNBC expressing the highest level of *PRMT1* mRNA (most proliferative) display better RFS, possibly because they respond better to chemotherapy, as observed for other targets linked to proliferation [30,31].

At the protein level, PRMT1 is more expressed in BC compared to normal tissues, confirming previous reports [37,44]. Here, we accounted for BC heterogeneity and found that PRMT1 protein is expressed at similar levels in the different BC subtypes. We observed both nuclear and cytosolic staining for PRMT1 which is in apparent contrast to a study showing mainly cytosolic localization [44], using an antibody that also recognizes the C-terminus of PRMT1, thereby detecting all its isoforms [45]. Several PRMT1 splice variants have been described which show cytoplasmic and/or nuclear localization [43]; therefore, it may not be surprising to detect PRMT1 in both compartments. Furthermore, PRMT1 is a well-described regulator of transcription, by methylating histones and transcription factors [14]. PRMT1 interacts with the progesterone receptor in the nucleus of breast cancer cells [39]. In addition, PRMT1 is expressed in both the cytosol and the nucleus in renal [46,47] and pancreatic [48] carcinomas. We also detected PRMT1 at the plasma membrane, preferentially in the ER-negative BC subtypes, possibly since it interacts with some transmembrane receptors such as EGFR [20,21] or IGF-1R [49]. However, we cannot exclude that the PRMT1 antibody we used recognizes the plasma membrane-associated PRMT8, although it is brain-specific, as it shares 80% homology with PRMT1.

Transcriptomic analysis highlighted several pathways regulated by PRMT1. Here, we focused on two pathways that are known to be activated in TNBC [3,4,5]. PRMT1 has been previously observed to modulate EGFR signaling by two mechanisms: (i) by methylating histone H4 (H4R3me2a) on its promoter in colorectal cancer (CRC) [23] and glioblastoma cells [24] and (ii) by methylating EGFR in CRC and TNBC cells [20,21]. Here, we demonstrate that PRMT1 itself is directly recruited to the promoter of *EGFR*, thus activating its transcription.

The role of PRMT1 on Wnt signaling is ambiguous since PRMT1 can be both an activator and an inhibitor of this pathway. On the one hand, PRMT1 can inhibit Wnt signaling by methylating two antagonists (i) Axin (in HEK293 and L929 cell lines) [27] and (ii) Dishevelled (in HEK293, B2b, and F9 cell lines) [28]. On the other hand, PRMT1 can activate the Wnt signaling pathway by methylating two Dishevelled-binding components: G3BP1 (in F9 cells) [26] and G3BP2 (in F9, HEK293 and SW380 cells) [25]. Therefore, the role of PRMT1 on Wnt signaling may be context dependent. Here, we show that PRMT1 regulates the Wnt signaling pathway at the transcriptomic level. Indeed, we found that PRMT1 activates the transcription of two main components of the Wnt pathway—*LRP5* and *PORCN*—by being recruited to their promoter regions. Furthermore, we demonstrate that PRMT1 activates the canonical Wnt signaling pathway. Additionally, PRMT1 enzymatic activity could be required as type I PRMT inhibitors reduce Wnt signaling pathway. Hence, PRMT1 could activate the pathway by directly methylating Wnt components or methylating histones on their promoters. Together, this implies that PRMT1 may regulate the Wnt signaling pathway by regulating the amounts of LRP5 available at the plasma membrane and by controlling the Porcupine-dependent post-translational modification of Wnt ligands, which is required for their secretion.

As PRMT1 is highly expressed in BC, we evaluated its potential as a therapeutic target. We found that PRMT1 depletion (i) decreased the cell viability, (ii) blocked their clonogenic potential, and (iii) induced DNA damage and apoptosis in various cell lines of different BC subtypes. This is in accordance with previous reports in TNBC [21,37,50,51] and luminal [39,51,52] BC cell lines as well as cell lines of other cancer types [23,46,53,54,55]. We next addressed the question whether the enzymatic activity of PRMT1 was required for BC cell survival. To date, there are no PRMT1 specific small-molecule inhibitors, but rather inhibitors that target all type I PRMTs, with some selectivity towards PRMT1, PRMT6, and PRMT8 [18,19]. GSK3368715 targets these three PRMTs at similar IC_50_ [19], whereas PRMT6 and PRMT8 are more sensitive than PRMT1 to MS023 [18]. We observed differential sensitivity among BC cell lines to both type I PRMT inhibitors, suggesting the need to identify biomarkers of response. This may perhaps help stratify patients who could benefit from treatment with these type I PRMT inhibitors. Together, we found that PRMT1 and its enzymatic activity are required for BC cell survival; however, we cannot rule out the influence of PRMT6 activity when using these inhibitors in our BC cell lines.

When assessing these inhibitors in combination with chemotherapies used in the clinic to treat TNBC patients, we observed synergistic interactions with cisplatin, cyclophosphamide, and camptothecin, but not with docetaxel and paclitaxel in MDA-MB-468 cells. Notably, these synergistic interactions occurred at doses lower than the IC_50_ of each drug, therefore potentially minimizing their cytotoxic side-effects when used in combination in vivo. MS023 treatment was shown to sensitize ovarian cancer cells to cisplatin [56] and CRC cells to SN-38, a camptothecin derivative [57]. In order to generalize our findings, we are currently evaluating these combinations in additional TNBC cell lines.

The highest synergy was observed when we combined both type I PRMT inhibitors with erlotinib in MDA-MB-468 cells, a cell line overexpressing EGFR [17]. It would be valuable to test this combination in other TNBC cell lines to verify whether this synergy is associated with EGFR overexpression. We have previously reported a synergistic interaction between erlotinib and a PRMT5 inhibitor, independently of the EGFR expression status of TNBC cell lines [17]. Although EGFR is overexpressed in TNBC, targeting EGFR on its own has shown only a modest effect in clinical trials in TNBC patients [3]. Considering our results, it may be beneficial to combine EGFR and PRMT inhibitors to treat TNBC. However, this hypothesis must be tested *in vivo* in various TNBC patient-derived xenograft (PDX) models. Additional studies have reported that type I PRMT inhibitors synergize with inhibitors targeting PARP in TNBC [58] and lung cancer [59]; PRMT5 in leukemia, pancreatic, and lung cancer [19,60,61]; FLT3 kinase in leukemia [62,63]; or anti-PD-1/PD-L1 in various cancer types [64,65]. Altogether, this also highlights the potential clinical relevance of combining type I PRMT inhibitors with targeted therapies.

We performed pre-clinical studies to explore the translational relevance of targeting PRMT1 using GSK3368715, which is being evaluated in a phase I clinical trial. We show that this inhibitor significantly reduced tumor growth in an MDA-MB-468-derived xenograft model, aligning with a previous study (supplemental data from [19]). In contrast to Fedoriw et al., who directly injected these cells into the mice [19], we employed a two-step protocol in order to engraft tumors before treating the mice to better represent the clinical setting. In this condition, we observed a similar reduction in tumor growth by using a reduced inhibitor dose (80 mg/kg in our study vs. 150 mg/kg [19]). Type I PRMT inhibitors have also been shown to decrease tumor growth in other cancer types such as lymphoma [19], pancreatic [19,38], hepatocellular carcinoma [66], and colon [64,67] cancers. Therefore, targeting type I PRMTs could represent a new treatment strategy in various cancer types, including BC. Additionally, we have evidence supporting the idea that combining the type I PRMT inhibitors with chemotherapies or targeted therapies could be beneficial for the treatment of TNBC. This must be evaluated in various TNBC PDX models to account for the inter- and intra-tumor heterogeneity observed within TNBC [2]. Intra-tumor heterogeneity poses a major challenge in treating TNBC patients because of a subpopulation of cells resistant to chemotherapies, leading to residual disease and relapse [2]. These chemo-resistant cells are believed to be fueled by developmental pathways such as the Wnt signaling pathway [2,4,5], hence, inhibiting PRMT1 may eradicate these resistant cells. Therefore, addressing whether the drug combinations identified here (*in vitro*) could overcome relapse in chemo-resistant TNBC PDX models would be clinically valuable.

## 5. Conclusions

The current paucity of targeted therapies for TNBC patients has prompted researchers to find novel treatment strategies. PRMT enzymes have recently emerged as attractive therapeutic targets for several cancer types, including BC. Here, we report that PRMT1, the major type I PRMT, is highly expressed in all BC subtypes, regulates two major signaling pathways activated in TNBC (EGFR and Wnt), and is required for cell survival. In addition, our study suggests that the combinatorial inhibition of type I PRMTs with chemotherapies could be clinically beneficial for TNBC patients.

## Figures and Tables

**Figure 1 cancers-14-00306-f001:**
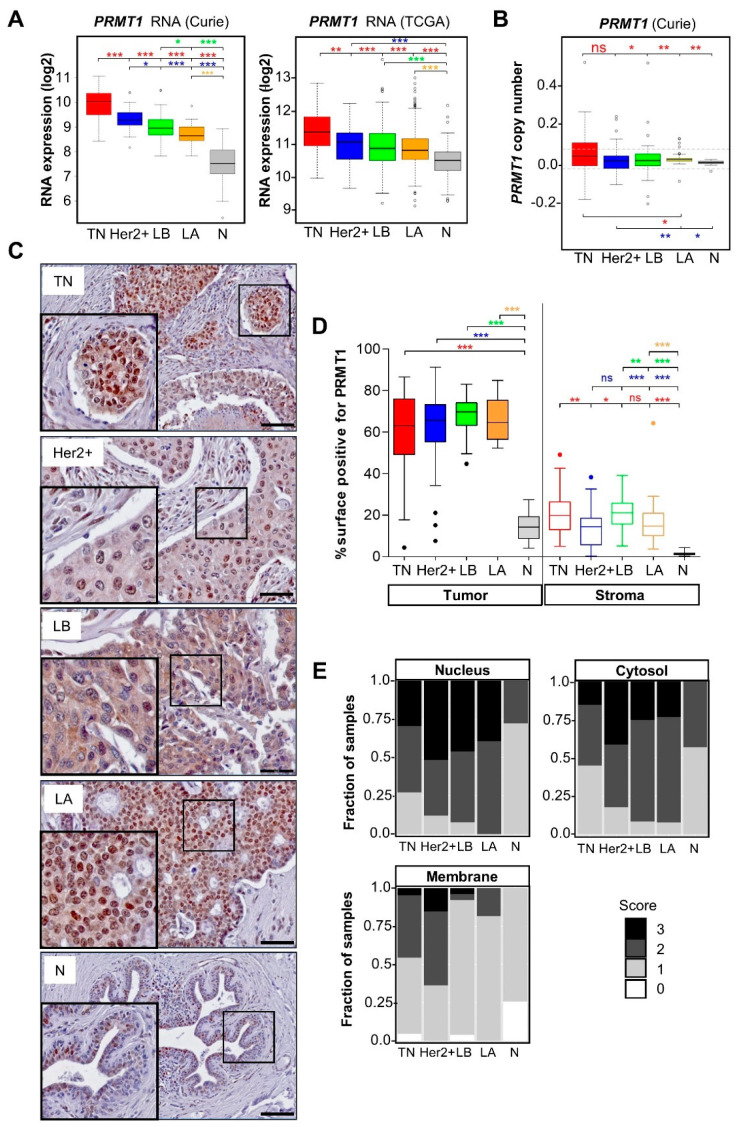
PRMT1 is highly expressed in breast tumors. (**A**) High levels of *PRMT1* mRNA in breast cancer. PRMT1 RNA expression in TNBC (TN, red), Her2+ (blue), Luminal B (LB, green), Luminal A (LA, orange), and healthy breast tissues (N, grey) in Curie (**left panel**) and TCGA (**right panel**) cohorts is illustrated by box plots (log2 transformed). (**B**) High *PRMT1* DNA copy number (CN) in TNBC in the Curie cohort. *PRMT1* DNA CN determined by Affymetrix microarray analysis is presented in boxplots (smoothed segmented CN signal), with dashed lines indicating the thresholds retained to call CN gains and losses (see Appendix A for the number of samples showing loss or gains). (**C**) High levels of PRMT1 protein in BC. PRMT1 protein levels were analyzed by IHC in the Curie cohort. A representative image of PRMT1 staining is shown for the different BC subtypes (scale bar = 50 μM). (**D**) Quantification of the tumoral (**left**) or stromal (**right**) surface positive for PRMT1 staining represented as a percentage compared to the total surface. Open and closed circles represent outlier tumors within the different populations (A,B,D). (**E**) Intensity scores of PRMT1 staining in the different cellular compartments (0: no staining, 3: the strongest staining). * *p* < 0.05; ** *p* < 0.01; *** *p* < 0.001; ns = not significant, as calculated using the Student *t*-test (**A**), Fischer exact test (**B**) or Mann–Whitney test (**D**).

**Figure 2 cancers-14-00306-f002:**
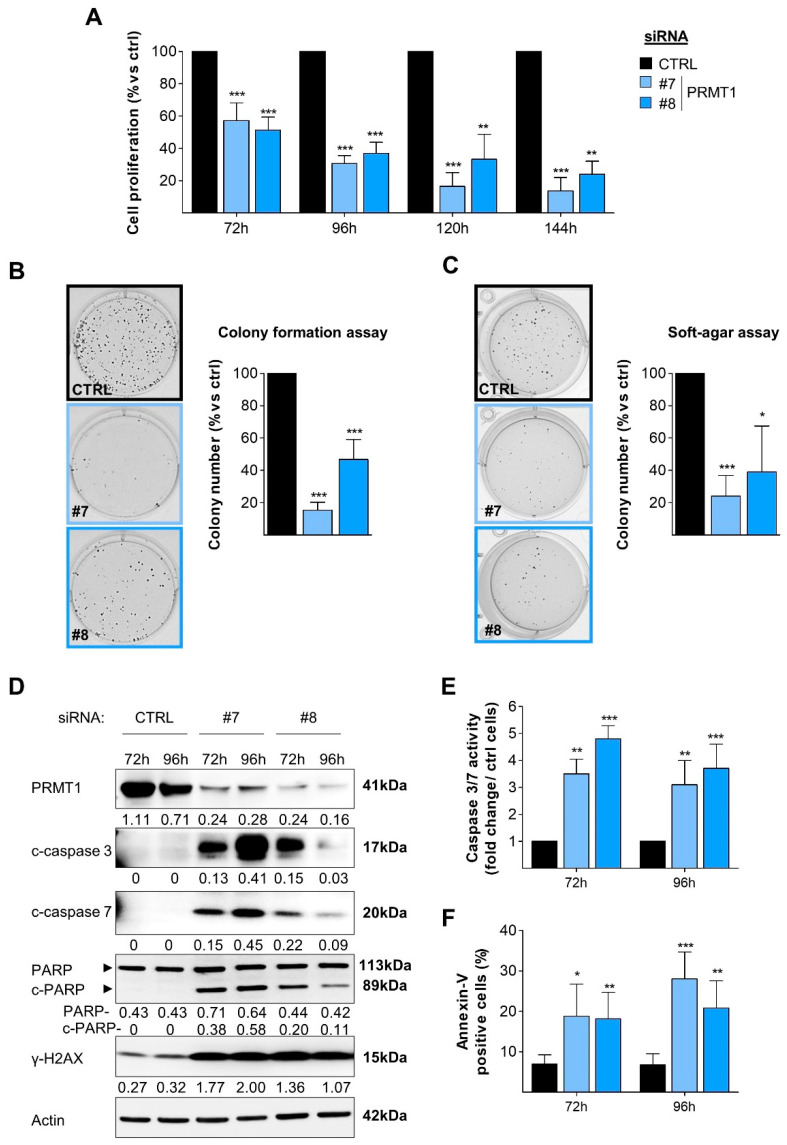
PRMT1 depletion decreases cell viability and induces apoptosis of MDA-MB-468 cells. (**A**) PRMT1 depletion impairs cell viability (MTT assay). Cells were transfected with control (CTRL, black) or two different PRMT1 siRNAs (#7, #8, blue) for 72–144 h. (**B**,**C**) PRMT1 depletion impairs colony formation when cells are grown on plastic for 13 days (**B**) or in soft agar for 4 weeks (**C**) following siRNA treatment. (**D**–**F**) PRMT1 depletion induces apoptosis. Apoptosis was detected by Western blotting using antibodies recognizing the cleaved forms of caspase 7 (c-caspase 7), caspase 3 (c-caspase 3) and PARP (c-PARP) (**D**), by caspase 3/7 assay (**E**) or annexin-V staining (**F**) after 72 h and 96 h following siRNA treatment. DNA damage was detected using an anti-γH2AX antibody (**D**). PRMT1 depletion was verified using an anti-PRMT1 antibody (**D**). Anti-actin antibody was used as a loading control and quantification of the bands (normalized to the loading control) are indicated below each blot (**D**). Results are presented as the percentage (**A**–**C**,**F**) or fold change (**E**) relative to control cells (CTRL). For the quantifications, the data are expressed as the mean ± SD from at least three independent experiments (**A**–**C**,**E**,**F**). Pictures are from a single experiment, representative of three independent experiments (**B**–**D**). *p*-values are calculated from a Student *t*-test and represented as * *p* < 0.05; ** *p* < 0.01; *** *p* < 0.001 (**D**).

**Figure 3 cancers-14-00306-f003:**
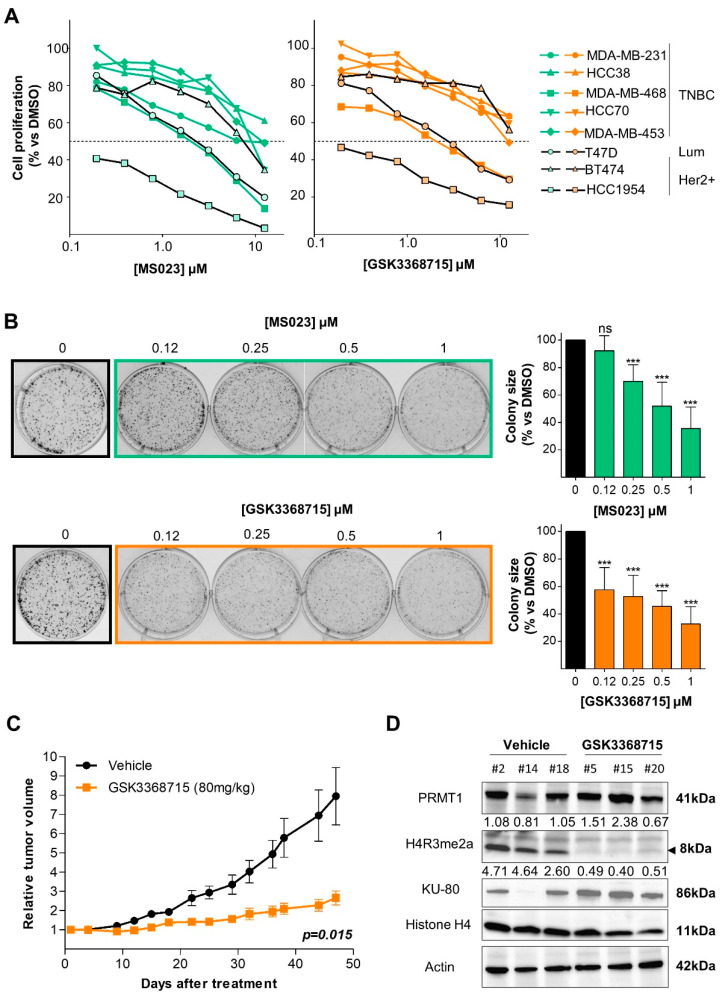
Type I PRMT inhibitors reduce cell viability and tumor growth. (**A**) Type I PRMT inhibitors decrease BC cell viability. TNBC, luminal (Lum), and Her2+ cells were treated with MS023 (**left panel**) or GSK3368715 (**right panel**) for 7 days, except MDA-MB-231 (4 days), and proliferation was determined by MTT or WST1 assays. Results are presented as the average percentage of cell growth relative to DMSO-treated cells from three independent experiments. (**B**) Type I PRMT inhibitors reduce the growth of colonies when MDA-MB-468 cells were cultured on plastic for 9 days after MS023 (**top**) or GSK3368715 (**bottom**) treatment. Quantification of colony size is expressed as a percentage relative to DMSO-treated cells, represented as the mean ± SD from at least three independent experiments (**right panel**). Pictures are from a single experiment representative of three independent experiments (**left panel**). *p*-values are from a Student *t*-test and represented as *** *p* < 0.001; ns = not significant. C, GSK3368715 slows tumor growth. Tumors derived from MDA-MB-468 cells were subcutaneously grafted into 12 mice (6 vehicle-treated, black; 6 GSK3368715-treated, orange). Growth curves were obtained by plotting mean relative tumor volume ± SEM as a function of time. *p*-value was calculated using a Mann–Whitney U test. (**D**) GSK3368715 inhibits PRMT1 activity *in vivo*. PRMT1 expression (anti-PRMT1) and activity (anti-H4R3me2a) were analyzed in the tumors excised from 3 vehicle (#2, #14, #18) or GSK3368715 (#5, #15, #20)-treated mice at the end of the experiment (**C**). Antibodies against histone H4, actin and KU-80 were used as controls and quantification of the bands (normalized to the actin band) are indicated below each blot.

**Figure 4 cancers-14-00306-f004:**
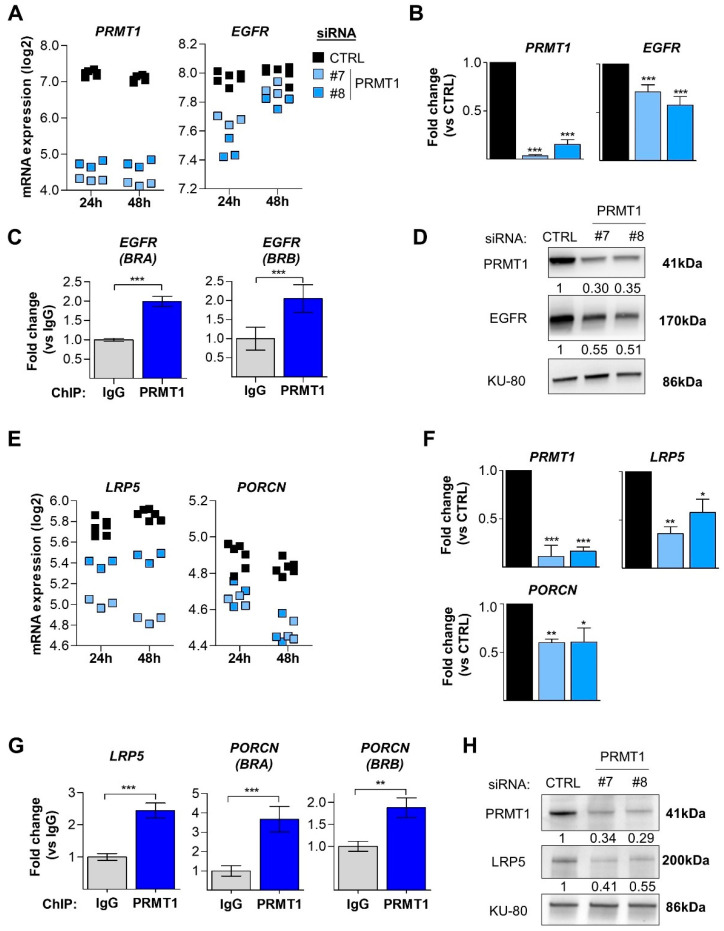
PRMT1 is enriched on the promoter of *EGFR*, *LRP5* and *PORCN* and activates their transcription. (**A**,**B**) PRMT1 depletion in MDA-MB-468 cells reduces *EGFR* mRNA expression as shown by Affymetrix microarray (**A**) and verified by qPCR (**B**). (**C**) PRMT1 is recruited to two promoter regions (Binding Region A, BRA; BRB) of *EGFR*. (**D**) PRMT1 depletion reduces EGFR protein level as shown by Western blotting. (**E**,**F**) PRMT1 depletion reduces *LRP5* and *PORCN* mRNA expression as shown by Affymetrix microarray (**E**) and validated by qPCR (**F**). (**G**) PRMT1 is recruited to the promoter of *LRP5* and two promoter regions (BRA, BRB) of *PORCN*. (**H**) PRMT1 depletion reduces LRP5 protein level as shown by Western blotting. MDA-MB-468 cells were transfected with control (black) or two PRMT1 (#7, #8, blue) siRNAs for 24 h (**A**,**B**,**E**) and 48 h (**A**,**D**–**F**,**H**). mRNA expression was logarithmically transformed (log 2) and each replicate is represented as a single point on the scatter plot (**A**,**E**). ChIP experiments were performed using anti-PRMT1 (blue bars) or anti-IgG (grey bars) antibodies using chromatin isolated from MDA-MB-468 cells (**C**,**G**). qPCR was performed using primers targeting the promoter regions of *EGFR* (**C**), *LRP5* (**G**) and *PORCN* (**G**). PRMT1 depletion was verified in the Affymetrix microarray (**A**), by qPCR (**B**,**F**) and by Western blotting (**D**,**H**). Antibody against KU-80 was used as a loading control for the Western blots and pictures are representative of at least three independent experiments (**D**,**H**). Intensity ratios of the bands, indicated below each blot, represent a fold change relative to control siRNA, after normalization to the loading control (**D**,**H**). The quantifications are represented as a fold change relative to the control siRNA (**B**,**F**) or control IgG (**C**,**G**) and presented as mean ± SD (**B**,**F**) or mean ± SEM (**C**,**G**) from three independent experiments. *p*-values from Student *t*-test are represented as * *p* < 0.05; ** *p* < 0.01; *** *p* < 0.001.

**Figure 5 cancers-14-00306-f005:**
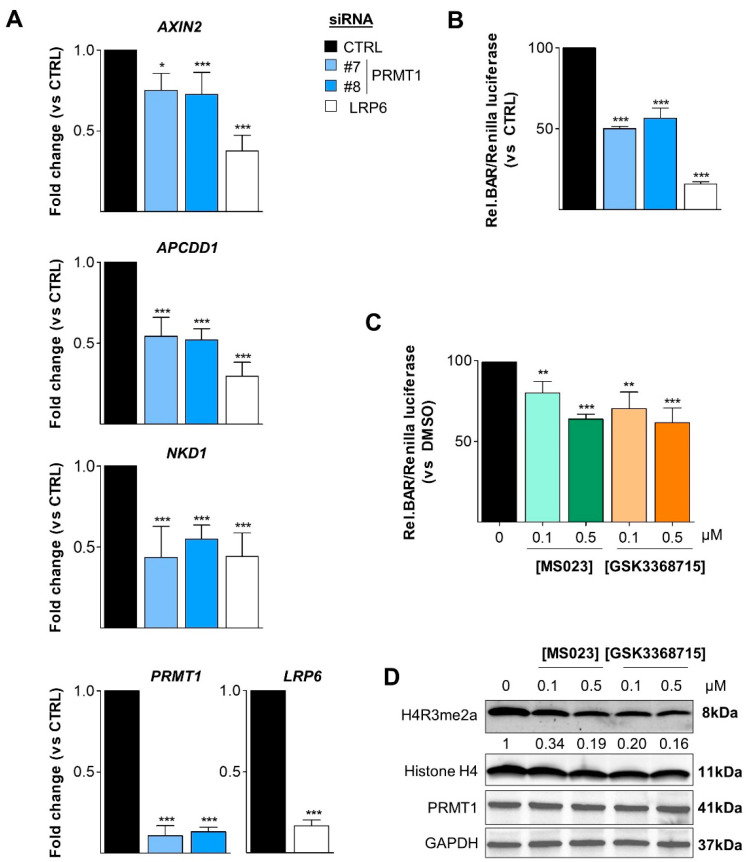
PRMT1 activates the canonical Wnt signaling pathway. (**A**,**B**) PRMT1 depletion decreases Wnt signaling activity. MDA-MB-468 cells were transfected with control (CTRL, black), two PRMT1 (#7, #8, blue) or LRP6 (white) siRNA for 48 h (**A**,**B**), and then co-transfected with plasmids coding for BAR-firefly luciferase and Renilla luciferase for 24 h (**B**), before Wnt3a stimulation for 6 h (**A**,**B**). The expression of *AXIN2*, *APCDD1*, *NKD1* (Wnt target genes), *PRMT1* and *LRP6* were quantified by qPCR (normalized to actin) (**A**). The relative luciferase signal (firefly luciferase/Renilla luciferase) is represented as a percentage normalized to the control siRNA (CTRL) (**B**). siRNA targeting LRP6 was used as a positive control (**A**,**B**). (**C**) Type I PRMT inhibitors decrease Wnt signaling activity. MDA-MB-468 cells were treated with 0.1 μM or 0.5 μM of MS023 (green) or GSK3368715 (orange) for 48 h, and then co-transfected with plasmids coding for BAR-firefly luciferase and Renilla luciferase for 24 h, before Wnt3a stimulation for 6 h. The relative luciferase signal (firefly luciferase/Renilla luciferase) is represented as a percentage normalized to the DMSO-treated cells (black). (**D**) PRMT1 inhibition was verified in this experiment (**C**) by Western blotting using anti-H4R3me2a antibody. Anti-histone H4, PRMT1, and GAPDH were used as loading controls. Intensity ratio of methylated histone H4 is indicated as a fold change relative to DMSO, after normalization to the loading control (**D**). All quantifications are represented as a fold change (**A**) or percentage (**B**,**C**) relative to the control. The data are expressed as the mean ± SD from at least three independent experiments (**A**–**C**). *p*-values from Student *t*-test are represented as * *p* < 0.05; ** *p* < 0.01; *** *p* < 0.001.

**Figure 6 cancers-14-00306-f006:**
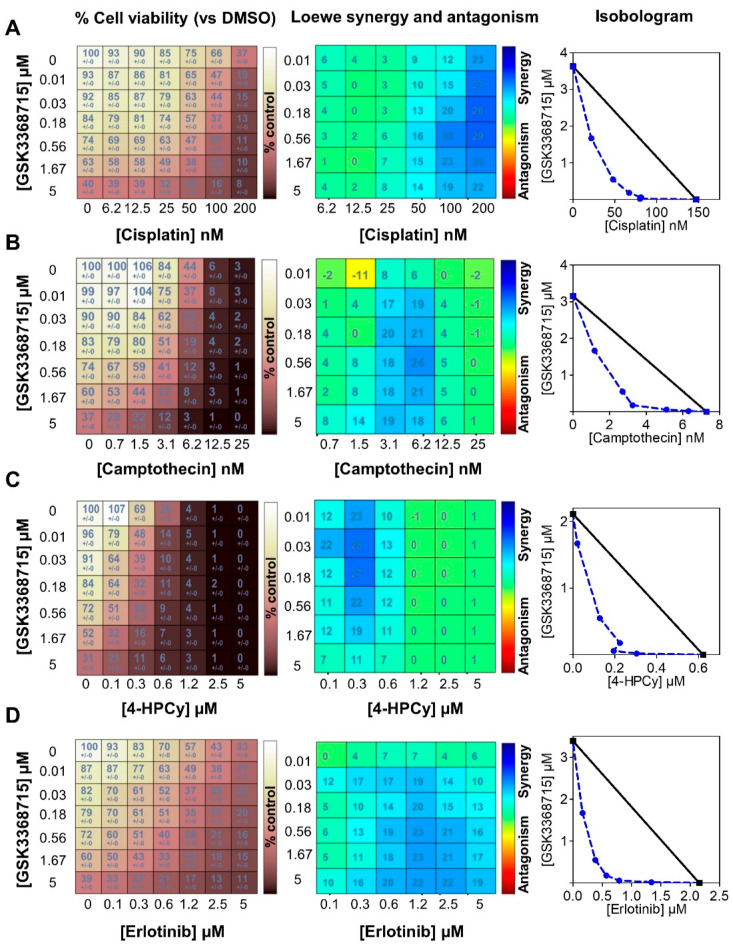
Synergistic interactions between GSK3368715 (a type I PRMT inhibitor) and chemotherapies (**A**–**C**) or erlotinib (**D**). MDA-MB-468 cells were seeded in 96-well plates, treated with the indicated drugs for 7 days (equivalent to four doubling times), and cell viability was measured by CellTiterGlo assay. GSK3368715 was serially diluted three-fold and cisplatin (**A**), camptothecin (**B**), 4-hydroperoxy cyclophosphamide (4-HPCy; (**C**)), erlotinib (**D**) were serially diluted two-fold (concentrations indicated in the figure). The drug interactions were calculated using the Loewe model on the Combenefit software. Cell viability (% compared to DMSO-treated cells, **left panel**), synergy matrix as calculated using the Loewe excess model (**middle panel**), and isobolograms (**right panel**) for each drug pair are indicated. Presented data are representative of three independent experiments.

## Data Availability

The transcriptomic data generated in this study are available in supplementary data files (Appendix A).

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
