# Peer review of "PRMT1 Regulates EGFR and Wnt Signaling Pathways and Is a Promising Target for Combinatorial Treatment of Breast Cancer"

_cancers, 2022, doi:10.3390/cancers14020306_

Round 1
Reviewer 1 Report
In this manuscript Suresh et al demonstrate that PRMT1 is highly expressed in breast cancer patients with the highest expression seen in triple negative breast cancer patients. Further, they show that knock-down of PRMT1 in various breast cancer cell lines decreases proliferation and anchorage dependent growth whereas increases DNA damage and apoptosis. Notably, the authors also show that PRMT1 specific inhibitors also decrease cancer cell viability in a dose dependent manner without affecting PRMT1 levels. Using microarray analysis, the authors further show that KD of PRMT1 decreases EGFR levels as well as Wnt signaling pathway in breast cancer cells by directly binding to the promoters of EGFR and WNT pathway genes. Furthermore, the authors demonstrate that PRMT1 inhibitors synergistically reduce cancer cell viability in combination with various chemotherapeutic drugs or EGFR inhibitors. Overall, the findings are promising and could lead to development of combination therapies for the most aggressive breast cancers. There are some minor concerns that the authors need to address:
- In Supplementary Figure S2A, the molecular weight mentioned for PRMT1 is not correct. It should be 41 kDa for PRMT1 and not 8 kDa.
- Also the molecular weight for actin is wrongly mentioned as 37 kDa in lot of figures instead of 42 kDa (eg Figure 2D, 3D, S1D, S2D among others). Please make sure that the molecular weights of other proteins are also mentioned correctly.
Reviewer 2 Report
To the authors:
This study aims to identify new therapeutic strategies for triple-negative breast cancer, highlighting the role of protein arginine methyltransferase 1 (PRMT1) in BC subtypes, specifically in TNBC. Furthermore, this study seeks to clarify whether this enzyme could represent an attractive therapeutic target for several types of cancer, including breast cancer, in combination with other currently used treatments. In general, TNBC and breast cancer have one of the worst prognoses due to their inter- and intra-tumoral heterogeneity leading to chemotherapy resistance and relapse. Therefore, studies such as the present one could provide valuable information on how to better understand the disease and develop new strategies to improve the survival of TNBC patients. However, despite the significant amount of work done, some weaknesses need to be addressed before publication.
General Comments:
In general, the manuscript is well written and reads easily. The experiments appear to be sound, thoroughly well and planned and carried out.
Introduction: The introduction section describes the main pathways involve in BC, including the current treatments that are used, describing PRMT proteins as novel targets. This section is well structured, describing clearly all this information, but in order to clarify the main objective of this study, the authors should add a sentence at the end explaining it at the end of this section.
Methods: It is well structured and the sample is big enough to consider conclusive results, highlighting that one strong point in this study is the composition of the cohort, because the authors have included the four types of breast cancer. I have only one question concerning the drugs/inhibitors concentrations which have been used. How did the authors decide the range of concentrations used in the drug combinations, and Could the authors add references if these data were obtained from published studies?
Results: The results are clearly explained. Nevertheless, However, I have some doubts.
- I would like to know how the authors have quantified PRMT1 staining in the different cell compartments shown in Figure 1E. How have they included all the cell lines studied?
- In relation to this, and given that the authors say in the discussion that “We observed both nuclear and cytosolic staining for PRMT1 which is in apparent contrast to a study showing mainly cytosolic localization” (P18L477), it would be interesting to perform an experiment after treating the cells with some EGFR ligand to obtain cytosolic and nuclear lysates and by western blot to see if there is a translocation from the cytoplasm to the nucleus of PRMT1, which would support the idea of its activity as a nuclear transcription factor. These data would corroborate those obtained by immunohistochemistry.
- Based on their results, the authors say “PRMT1 regulates the expression of EGFR, LRP5 and PORCN by directly binding to their promoter regions”(P12L374). Did the authors check if there are some common patterns in the promoter sequences of EGFR, LRP5 and PORCN that could confirm this idea?
Discussion: The authors say that “TNBC expressing the highest level of PRMT1 (most proliferative) display better RFS, most likely because they respond better to chemotherapy”(P18L473). But, are there any studies that support this theory? If it is the case, the reference should be added.
As I said in results, the authors say that “we found that PRMT1 activates the transcription of two main components of the Wnt pathway: LRP5 and PORCN, by binding to their promoters”(P18L502). Have any promoter sequences already been described for these genes to support this idea?
Finally, I would like to congratulate the authors for all the complementary data, which are very comprehensive.
Reviewer 3 Report
The manuscript by Suresh et al. provides interesting results as to the role played by PRMT1 in the progression of breast cancer. The results indicate that the oncogenic role played by PRMT in breast cancer involves activation of the EGFR and WNT signaling pathways. Finally targeting PRMT1 with appropriate inhibitors is likely to represent a useful strategy for the treatment of breast cancer with particular reference to TNBC tumors. Overall the study is well conceived and presented in a clear manner.
Minor points
1. The english language should be checked as, we noticed a few imperfections throughout the text.
2. The following portion of the Introduction: "...Asymmetric dimethylation is carried out by Type I PRMTs (PRMT1-4, 62 PRMT6, and PRMT8) with PRMT1 being responsible for most of this modification [8-14]. PRMTs are ubiquitously expressed, apart from PRMT8 which is brain-specific [12]...." should be rewritten, as it is unclear and difficult to understand.
